# Health-Related Quality of Life in Locally Advanced Gastric Cancer: A Systematic Review

**DOI:** 10.3390/cancers13235934

**Published:** 2021-11-25

**Authors:** Romy M. van Amelsfoort, Karen van der Sluis, Winnie Schats, Edwin P. M. Jansen, Johanna W. van Sandick, Marcel Verheij, Iris Walraven

**Affiliations:** 1Department of Radiation Oncology, Netherlands Cancer Institute, 1066 CX Amsterdam, The Netherlands; r.v.amelsfoort@nki.nl (R.M.v.A.); epm.jansen@nki.nl (E.P.M.J.); marcel.verheij@radboudumc.nl (M.V.); 2Department of Surgical Oncology, Netherlands Cancer Institute, 1066 CX Amsterdam, The Netherlands; k.vd.sluis@nki.nl (K.v.d.S.); j.v.sandick@nki.nl (J.W.v.S.); 3Scientific Information Service, Netherlands Cancer Institute, 1066 CX Amsterdam, The Netherlands; w.schats@nki.nl; 4Department of Radiation Oncology, Radboud University Medical Center, 6525 GA Nijmegen, The Netherlands; 5Department for Health Evidence, Radboud University Medical Center, 6525 GA Nijmegen, The Netherlands

**Keywords:** health-related quality of life, gastric cancer, curative treatment

## Abstract

**Simple Summary:**

Evaluating health-related quality of life (HRQOL) is important because it reflects the impact of treatment from the patient’s perspective. This holds especially true when prognosis is poor, as is the case for gastric cancer patients. This systematic review was performed to evaluate the short- and long-term impacts of current treatments on HRQOL in patients with locally advanced gastric cancer. The results of this review show that surgery and chemoradiotherapy have a significant impact on short-term HRQOL, although this recovers after 6–12 months. More focus on HRQOL in current clinical practice might improve patient counselling and monitoring before, during, and after treatment in locally advanced gastric cancer patients.

**Abstract:**

Background: Current treatment strategies have been designed to improve survival in locally advanced gastric cancer patients. Besides its impact on survival, treatment also affects health-related quality of life (HRQOL), but an overview of reported studies is currently lacking. The aim of this systematic review was therefore to determine the short- and long-term impact of chemotherapy, surgery, and (chemo)radiotherapy on HRQOL in locally advanced, non-metastatic gastric cancer patients. Methods: A systematic review was performed including studies published between January 2000 and February 2021. We extracted studies published in Medline, Embase, and Scopus databases that assessed HRQOL in patients with locally advanced, non-metastatic gastric cancer treated with curative intent. Studies using non-validated HRQOL questionnaires were excluded. Short-term and long-term HRQOL were defined as HRQOL scores within and beyond 6 months after treatment, respectively. Results: Initially, we identified 8705 articles (4037 of which were duplicates, i.e., 46%) and ultimately included 10 articles. Most studies reported that short-term HRQOL worsened in the follow-up period from 6 weeks to 3 months after surgery. However, recovery of HRQOL to preoperative levels occurred after 6 months. After completion of chemoradiotherapy, the same pattern was seen with worse HRQOL after treatment and a recovery of HRQOL after 6–12 months. Conclusions: In patients with locally advanced, non-metastatic gastric cancer, HRQOL deteriorated during the first 3 months after surgery and chemoradiotherapy. However, the long-term data showed a recovery of HRQOL after 6–12 months. To implement HRQOL in clinical decision making in current clinical practice, more research is needed.

## 1. Introduction

As the fourth cause of cancer-related mortality worldwide, gastric cancer remains a major health problem [1]. Despite intensive treatment, the prognosis remains poor, and the majority of Western patients recur within two years after treatment [2,3,4,5,6]. Current curative treatment options for locally advanced gastric cancer include surgery with postoperative chemoradiotherapy (adjuvant leucovorin + fluorouracil and 45 Gray) based on the Intergroup 0116 trial and perioperative chemotherapy based on the MAGIC trial (epirubicin + cisplatin + fluorouracil) and FLOT4 (fluorouracil + leucovorin + oxaliplatin + docetaxel) trial in Western patients, while in Asia adjuvant chemotherapy is the preferred treatment [7,8,9,10,11,12]. In addition to improving survival, the potentially curative treatments also have an impact on the patient’s health-related quality of life (HRQOL). HRQOL is becoming increasingly important in clinical trials because it reflects the treatment impact from the patient’s perspective [13]. However, little is known about the impact of treatment on HRQOL in patients with locally advanced gastric cancer. Most studies evaluating HRQOL in gastric cancer patients focus on the impact of surgery alone. Deterioration of HRQOL after a gastrectomy has been described [14,15]. However, the impact on HRQOL of current curative treatment strategies is still unclear [14,15]. These considerations prompted us to conduct this systematic review. Besides the impact of treatment on HRQOL, recent studies in gastric cancer have shown that HRQOL data before treatment have prognostic value [16,17]. Worse HRQOL before treatment was associated with worse survival during follow-up in patients with (locally) advanced gastric cancer [16,17]. The importance of measuring HRQOL is evident, not only during but also prior to treatment. Therefore, HRQOL can also play an important role in clinical decision-making. To be able to inform patients on the course of HRQOL after treatment, we wanted to investigate the effects of treatment on both short- and long-term HRQOL.

The aim of this systematic review is to provide an overview of the available literature on HRQOL in patients treated for locally advanced, non-metastatic gastric cancer. We have therefore formulated the following research questions: (1) What is the short-term impact of chemotherapy, surgery, and (chemo)radiotherapy on HRQOL in patients with locally advanced, non-metastatic gastric cancer? (2) What is the long-term impact of treatment on HRQOL in patients with locally advanced, non-metastatic gastric cancer?

## 2. Materials and Methods

### 2.1. Study Protocol

We performed a systematic literature review which is registered in the PROSPERO database, an international database of prospectively registered systematic reviews (registration number: CRD42021249987). We used the PRISMA (Preferred Reporting Items for Systematic Reviews and Meta-Analyses) guidelines for our protocol and the systematic review [18].

### 2.2. Search Strategy

A systematic search of the Medline, Embase, and Scopus databases was performed on 12 February 2021. No additional studies have been included after this date. We searched for articles assessing HRQOL in patients with locally advanced, non-metastatic gastric cancer treated with curative intent. The full search strategy is described in the Appendix A.

### 2.3. Eligibility Criteria

Studies were included if they met the following inclusion criteria: (I) patients with locally advanced gastric cancer (Ib-IVa gastric according to the 8th TNM staging of the American Joint Committee on Cancer) treated with curative intent (surgical resection with (neo)adjuvant chemotherapy or adjuvant (chemo)radiotherapy); (II) the use of a validated HRQOL questionnaire; (III) article written in English language; and (IV) publishing date from the year 2000 and onwards.

In addition, if studies analyzed different stages of gastric cancer, only the locally advanced groups were used in the analysis.

Studies were excluded if: a non-validated questionnaire was used, when no original data was presented, reviews, study protocols, or abstracts with no full text available. Studies researching various stages of gastric cancer were excluded if the proportion of patients exceeded 20% stage Ia. This study investigated the impact of multimodal treatment in gastric cancer, whereas Ia is often treated less intensively.

### 2.4. Outcomes

Primary outcomes of the study were: (1) short-term impact of treatment on HRQOL; (2) long-term impact of treatment on HRQOL. Based on available literature, short-term HRQOL was defined as HRQOL scores within 6 months after treatment [19]. Long-term HRQOL was defined as HRQOL scores more than 6 months after treatment.

### 2.5. Data Extraction

All studies (titles, abstracts, and full texts) identified by the search were independently analyzed by two observers (R.v.A. and K.v.d.S.). Conflicting decisions were resolved by discussion and if persistent, screened by a third reviewer (I.W.). R.v.A. and K.v.d.S. extracted the following data of the included studies: author, number of patients included, patient characteristics (age, gender, and tumor stadium), study design, inclusion years, country, HRQOL questionnaires, time points of questionnaires, HRQOL subscales, and treatment strategy. Authors of articles who did not present all data in the original manuscript were contacted for additional results. Unfortunately, we did not receive sufficient data to perform a pooled analysis.

### 2.6. Risk of Bias Assessment

Risk of Bias was assessed and discussed by two observers (R.v.A. and K.v.d.S.). The revised Cochrane risk of bias tool (RoB2) was used to assess randomized trials [20]. Non-randomized trials were scored by the risk of bias in non-randomized trials of interventions (ROBINS-I tool) and the critical appraisal skills program (CASP) tool was used in cohort studies [21,22]. Additionally, a study population below 20 patients was considered to be at risk of bias based on sample size and therefore excluded from further analyses.

## 3. Results

### 3.1. Search

A flowchart of the included articles in this systematic review is displayed in Figure 1. Our search string yielded 8705 articles, of which 4037 duplicates were removed. The remaining articles were scored based on title and abstract, resulting in 276 articles being analyzed for full text. Furthermore, 261 articles were excluded based on incorrect stage of disease, only an abstract being available, use of a non-validated questionnaire, different tumor type, non-English text, inclusion before the year 2000, letter to the editor, insufficient data, and a study protocol (Figure 1). Data was extracted from the remaining 15 articles. Two articles (Goody et al., and Kassam et al.) reported the same study; therefore these were analyzed together [23,24]. We included the additional data we received from Avery et al. [25]. Four articles with a high risk of bias were excluded in the final analysis [26,27,28,29].

Baseline demographics are described in Table 1. Of the 10 included studies, the year of publication varied between 2010 and 2020. Most included studies investigated the impact of surgery on HRQOL (n = 8), followed by the impact of chemoradiotherapy (n = 2) and chemotherapy (n = 2) on HRQOL. There were no studies evaluating the impact of perioperative chemotherapy on HRQOL. Based on the risk of bias analysis, 4 of the 10 studies scored as having “some concerns” and 6 as having “low concerns”.

### 3.2. HRQOL Measures

The EORTC cancer-specific questionnaire (QLQ-C30) was used in the majority of the studies (n = 7, 70%), followed by the QLQ-STO22 (n = 4, 40%), EQ-5D (n = 1, 10%), FACT-G and FACT-Ga (n = 1, 10%), and PGSAS-45 (n = 1, 10%).

### 3.3. Impact of Surgery on HRQOL

This section included eight studies that examined the impact of surgery on HRQOL in gastric cancer patients. A summary of all the outcomes is displayed in Table 2.

Short-term HRQOL was impaired from 6 weeks to 3 months after surgery as described by Avery et al., and Munene et al. [25,30]. In a cohort study by Avery et al., HRQOL was evaluated in 58 patients undergoing potentially curative surgery. HRQOL was significantly worse 6 weeks after surgery compared to before surgery (except for cognitive and emotional functioning) [25]. Munene et al., reported similar results regarding the outcome of HRQOL. They evaluated HRQOL (FACT-G and FACT-Ga) in 43 patients treated in the majority with total gastrectomy and D2, D3 lymphadenectomy. HRQOL was deteriorated 3 months postoperatively [30]. However, in the long term, Avery et al., and Munene et al., did report a recovery of HRQOL to (almost) preoperative levels 6 months after surgery in [25,30]. Hereafter, HRQOL persisted until a recurrence was discovered or prior to death [25,30].

The long-term impact on HRQOL was investigated by Jakstaite et al., 6 to 18 months after gastric surgery. Almost half of the patients also received adjuvant chemotherapy. Patients aged 65 years and older had better global HRQOL scores and social functioning. No significant impact on HRQOL was seen based on gender, clinical stage, or adjuvant chemotherapy [31]. Kinami et al., also investigated long-term impacts on HRQOL (PGSAS-45) in patients who underwent a distal partial gastrectomy > 1 year ago. Locally advanced stage gastric cancer patients, in which the majority also received adjuvant chemotherapy before, showed similar HRQOL compared to early-stage gastric cancer (except for the dumping subscale which was found to be dependent on remnant stomach size) [32].

The impact of the type of surgery on HRQOL yielded mixed results. Munene et al., concluded that there was no difference in HRQOL between patients who underwent a partial or total gastrectomy [30]. Huang et al., did not describe a difference in HRQOL between a laparoscopic-assisted total gastrectomy compared to a totally laparoscopic total gastrectomy (except for the symptom subscales pain and dysphagia, which were worse in laparoscopic-assisted total gastrectomy) [33]. However, Brenkman et al., showed, in a multivariate analysis, that patients who received a distal gastrectomy or minimal invasive surgery had better HRQOL, especially on the gastrointestinal subscales [34]. Park et al., described worse HRQOL in patients who underwent a total gastrectomy compared to distal gastrectomy > 1 years after surgery [35].

Both Brenkman et al., and Xia et al., evaluated HRQOL in gastric cancer patients compared to the general population or healthy controls [34,36]. Brenkman et al., described an impaired HRQOL in most subscales in patients after a gastrectomy (> 1 year after treatment, compared to the general population) [34]. Xia et al., observed a similar association in which patients with gastric cancer had significantly worse HRQOL on all the subscales of the EQ-5D compared to healthy controls [36].

### 3.4. Impact of Postoperative Chemoradiotherapy on HRQOL

The impact of chemoradiotherapy on HRQOL was investigated in the following studies and summarized in Table 2.

Short-term HRQOL was worse after chemoradiotherapy as described by Zygogianni et al., and Goody/Kassam et al. [23,24,37]. Zygogianni et al., evaluated the short-term impact of three-dimensional multifield radiotherapy (n = 62) versus anteroposterior (AP/PA) radiotherapy (n = 35) on HRQOL based on the QLQ-C30 questionnaire. Patients who were treated with AP/PA radiotherapy had significantly worse global QOL, diarrhea, and loss of appetite compared to patients treated with multifield radiotherapy. Global HRQOL scores were nearly two times higher in the multifield radiotherapy group [37]. The short-term impact of chemoradiotherapy on HRQOL was also described by Goody/Kassam et al., and showed a similar trend as after surgery. After chemoradiation, HRQOL deteriorated compared to HRQOL before treatment. The long-term impact, at 6–12 months of follow-up, showed recovery of HRQOL. No significant differences were seen compared to HRQOL prior to chemoradiation [23,24].

### 3.5. Impact of Chemotherapy on HRQOL

The impact of chemotherapy on HRQOL in locally advanced gastric cancer was evaluated in the following studies (Table 2).

The short-term impact of chemotherapy on HRQOL was not evaluated in the included studies. The long-term impact on HRQOL was described by Jakstaite et al., and Brenkman et al. [31,34]. Adjuvant chemotherapy had no significant impact on HRQOL in the study of Jakstaite et al. [31]. Brenkman et al., described, based on a multivariate analysis, that patients who received neoadjuvant treatment had better HRQOL after gastrectomy on the following subscales: global QOL, nausea and vomiting, pain, dyspnea, diarrhea, reflux, and eating restrictions [34].

## 4. Discussion

The aim of this systematic review was to investigate the impact of chemotherapy, surgery, and (chemo)radiotherapy on short- and long-term HRQOL in locally advanced, non-metastatic gastric cancer patients. Our systematic review is the first systematic review investigating HRQOL in locally advanced gastric cancer patients. We found that HRQOL was impaired until 6 months of follow-up but from 6 months onwards, HRQOL recovered to pretreatment levels in most patients.

The impact of treatment is reflected as an impaired short-term HRQOL. Within studies researching the impact of surgery, a decline in physical, mental, and functioning scores were seen from 6 weeks to 3 months after surgery. This pattern was also observed after chemoradiotherapy. However, the long-term impact (after 6 months) showed a recovery of HRQOL for both surgery and chemoradiotherapy [23,24,25,30]. These results can play an important role in daily practice, particularly when discussing the expected dynamic course of HRQOL with patients and thereby clarify the expectations of the patient.

We observed that HRQOL in locally advanced gastric cancer patients was worse compared to healthy controls or the general population [34,36]. This discrepancy might be due to an already impaired HRQOL due to the cancer diagnosis at baseline. The prognostic role of pretreatment HRQOL was not addressed in this systematic review, but also has a potential added value as previously described [16,17]. This further emphasizes the importance of assessing baseline HRQOL.

Another interesting finding of this systematic review was that HRQOL deteriorates when disease recurs and prior to death [25,30]. This outcome is relevant, as HRQOL questionnaires can play a potential role in monitoring locally advanced gastric cancer patients. So, HRQOL can be important not only after treatment but also during follow-up in monitoring disease progression.

In this systematic review, the impact of the type of surgery on HRQOL is unclear, as contradictory conclusions were made. Two studies did not describe a difference between the type of surgery [30,33]. This outcome is in contrast to the results of Park et al., and Brenkman et al. [34,35]. Park et al., described worse long-term HRQOL in patients who underwent a total gastrectomy compared to distal gastrectomy, in both functioning (physical and role) as well as seven symptom scores (fatigue, pain, reflux, eating restrictions, anxiety, taste, and body image) [35]. Brenkman et al., also described a better HRQOL in patients undergoing distal gastrectomy, as well as treatment with minimal invasive surgery in a multivariate analysis [34].

The impact of chemotherapy is also under debate as Jakstaite et al., did not find an impact of adjuvant chemotherapy on HRQOL [31]. However, Brenkman et al., described, based on a multivariate analysis, that patients treated with neoadjuvant treatment had better HRQOL [34]. These results may differ due to a difference in HRQOL measured in time. Besides these contradictory results, too little is known about the short-term impact on HRQOL of (neo)adjuvant chemotherapy. Therefore, more research is needed to elucidate the effect of chemotherapy on both short -and long-term HRQOL.

Current treatment usually consists of perioperative chemotherapy. However, prospective studies that investigate the impact of perioperative chemotherapy on HRQOL in patients with locally advanced gastric cancer are currently non-existent. Hence, there is a great need for more research on this topic. Within the CRITICS study, HRQOL has been investigated as well. The CRITICS study compared perioperative chemotherapy to preoperative chemotherapy, surgery, and postoperative chemoradiotherapy [4]. The first results show that after postoperative chemo(radio)therapy, the chemotherapy group had significantly better physical functioning (*p* = 0.020, ES = 0.25) and less dysphagia (*p* = 0.010, ES = 0.35) compared to the chemoradiotherapy group [38].

Despite the small number of articles describing HRQOL in locally advanced gastric cancer, this study shows a trend in HRQOL similar to other cancers, e.g., esophageal cancer. Treatment of esophageal cancer (including chemoradiotherapy, surgery, and neoadjuvant therapy combined with surgery) also resulted in a short-term deterioration of HRQOL, specified as four months from the start of therapy [15]. Similar to the studies included in this systematic review, a statistically and clinically significant increase in fatigue was found in patients receiving chemoradiotherapy as well as a significant decrease in physical, role, and social functioning, and an increase in loss of appetite and diarrhea in patients receiving surgery. Within the long term, classified at 12 months, comparable results were found for the locally advanced gastric cancer patients. The functioning scores also recovered to baseline, whereas in patients receiving surgery, diarrhea remained. Patients receiving neoadjuvant therapy and surgery reported better short-term HRQOL, compared to other therapies; however, no differences in HRQOL were found compared to surgery alone in the long term. When long-term HRQOL in esophageal cancer patients was compared to healthy controls, HRQOL remained impaired, which is similar to the results of Brenkman et al., and Xia et al. [34,36,39]. The overall similarity between both cancers supports the trend shown in the HRQOL of locally advanced gastric cancer as presented in this study.

This systematic review has some limitations. First, there is a lack of high-quality studies investigating the quality of life in patients with locally advanced gastric cancer. Therefore, we only included a very small number of articles, hindering the possibility to perform a pooled analysis. We requested additional data, but unfortunately, we did not receive sufficient data to perform a pooled analysis. Second, there is also a difference in the quality of the included studies. Most studies are not randomized, and 4 of the 10 (40%) studies were scored as having “some concerns” in the Risk of Bias analysis [25,32,33,35].

A strength of this study is the focus on patients with locally advanced gastric cancer treated with curative intent, which is, to our knowledge, the first systematic review investigating this. Furthermore, we performed an extensive systematic review, including only studies published from the year 2000 and onwards. In addition, we only included studies that assessed HRQOL with validated questionnaires.

## 5. Conclusions

In conclusion, this is the first systematic review investigating HRQOL in locally advanced, non-metastatic gastric cancer patients. The impact that treatment has on HRQOL in locally advanced, non-metastatic gastric cancer worsened after surgery and chemoradiotherapy but recovered after 6–12 months. This dynamic course of HRQOL is of added value in current clinical practice, to clarify the patient’s expectations. More research is needed to implement HRQOL in clinical decision-making.

## Figures and Tables

**Figure 1 cancers-13-05934-f001:**
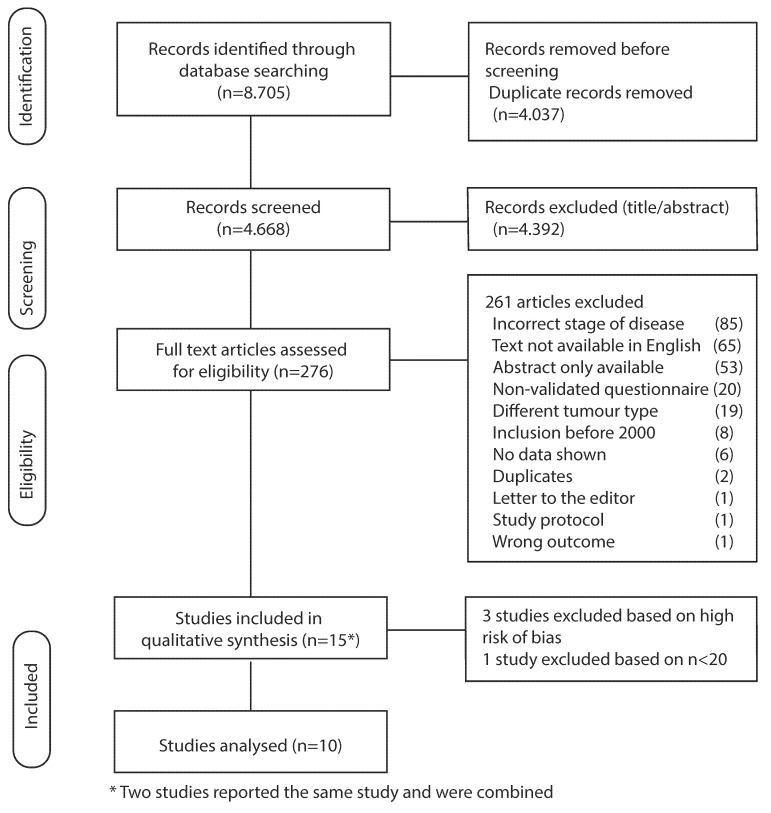
Flowchart of included studies.

**Table 1 cancers-13-05934-t001:** Included studies, baseline demographics.

Study	Study Design	Inclusion Years	Country	HRQOL Measures	Intervention	No. of Patients	Mean Age (Years)	Gender (Male)	Risk of Bias
Avery 2010	Cohort	2000–2004	United Kingdom	QLQ-C30, QLQ-STO22	HRQOL of patients with potentially curable gastric cancer	58	70 vs. 72	78%	Some concerns
Brenkman 2018	Cross-sectional	2001–2015	The Netherlands	QLQ-C30, QLQ-STO22	HRQOL after gastrectomy vs. reference population	222	68	66%	Low
Goody 2016/ Kassam 2010 *	Case-series	2002–2013/ 2022–2007	Canada	QLQ-C30	Radiotherapy + 12 weeks 5-FU + escalating doses cisplatin	55/33	54/56	55%	Low
Huang 2017	Cross-sectional	2014–2016	China	QLQ-C30, QLQ-STO22	Isoperistaltic anastomosis vs. Roux-en-Y anastomosis on HRQOL′	89	61.6	76%	Some concerns
Jakstaite 2012	Cross-sectional	2008–2009	Lithuania	QLQ-C30	HRQOL 6–18 months post-surgery (+ adjuvant chemotherapy)	34	64	59%	Low
Kinami 2020	Cross-sectional	2009–2014	Japan	PGSAS-45	Post-gastrectomy syndrome > 1 year post-partial gastrectomy. Early stages vs. advanced stage of gastric cancer	22	68.3	64%	Some concerns
Munene 2012	Cohort	2001–2007	Canada	FACT-G & FACT-Ga	HRQOL following gastrectomy	43	65	51%	Low
Park 2020	Cohort	2011–2014	Korea	QLQ-C30, QLQ-	HRQOL before and following total	300	59 vs. 63	70%	Some concerns
				STO22	gastrectomy and distal gastrectomy				
Xia 2020	Cross-sectional	2016–2017	China	EQ-5D	HRQOL in patients with gastric cancer	752	NR	75.2 vs. 68.7%	Low
Zygogianni 2018	Cross-sectional	2005–2014	Greece	QLQ-C30	HRQOL after completion of adjuvant radiotherapy: comparing anterior-posterior field vs. multifield technique	97	63	80%	Low

* = Same study, combined analysis; HRQOL = Health-related quality of life; NR = not reported.

**Table 2 cancers-13-05934-t002:** Summary of studies evaluating HRQOL.

Study	Aim study	Timepoints Questionnaires	Short-Term Impact on HRQOL	Long-Term Impact on HRQOL (>6 Months)
**Avery 2010**	HRQOL before and after potentially curative gastrectomy	Prior to surgery Post-surgery at: +6 weeks +3, +6, 9, 12, 18, 24 months	**Functioning:** Physical, role, and social functioning were impaired at 6 weeks post-surgery and started to recover at 3 months post-surgery. **Symptoms:** Increase reported in appetite loss, diarrhea, and eating restrictions.	**Functioning:** Global QOL, physical, role, and social functioning were recovered at 6 months. **Symptoms:** Diarrhea did not recover to baseline. Nausea/vomiting, diarrhea, pain, reflux, dry mouth, and sleep difficulties still reported in 50% of patients 6 months after surgery. Over 70% of patients reported fatigue and eating problems.
		(QLQ-C30, QLQ-STO22)		
**Brenkman 2018**	HRQOL after gastrectomy vs. reference population	Post-surgery (range 1 month–5 years)		**Functioning:** All functioning scores were impaired compared to the reference population (except for emotional functioning and global QOL). **Symptoms:** All symptom scores were impaired compared to the reference population (except for pain, insomnia, and constipation). **Multivariate analysis:** Patients undergoing neoadjuvant therapy, distal gastrectomy, or minimal invasive surgery had better HRQOL scores. Female gender was a predictive factor for nausea and vomiting, insomnia, appetite loss, diarrhea, body image, eating restrictions, and hair loss.
		(QLQ-C30, QLQ- STO22)		
**Huang 2017**	HRQOL after Isoperistaltic anastomosis (IJOM) vs. Roux-en-Y anastomosis after totally laparoscopic total gastrectomy	Post-surgery at: 6 months		**Functioning:** Functioning scores were comparable in both groups. **Symptoms:** Pain and dysphagia were experienced less in the IJOM group.
		(QLQ-C30, QLQ-STO22)		
**Jakstaite 2012**	HRQOL in relation to age, sex, clinical stage, postoperative complication, and adjuvant chemotherapy after total gastrectomy	Post-surgery (range 6–18 months)		**Functioning:** Role, emotional, social, and global QOL were worse in patients ≤ 65 years (clinically relevant *). A significant difference was observed in social functioning and global QOL. Role functioning was worse in male patients compared to female (clinically relevant *, ns). Global QOL, physical, role, cognitive, and social functioning scored worse in patients with a more advanced stage of disease (clinically relevant *, ns). **Symptoms:** Fatigue, nausea/vomiting, and insomnia were more common in patients ≤ 65 years and female patients (clinically relevant *, ns). Pain was more common in patients ≤ 65 years (clinically relevant *, ns). Constipation was more common in female patients, patients < 65 years, and patients with stage III (clinically relevant *, ns). Dyspnea, appetite loss, and diarrhea were more common in patients with lower stages of disease (clinically relevant *, ns).
		(QLQ-C30)		
**Goody 2016/** **Kassam 2010**	HRQOL after chemoradiation	Post-surgery After completion of radiotherapy After completion of chemotherapy at: +4 weeks, +6–12 months, +2–3 years	**Functioning:** Global QOL, role, and social functioning significantly declined after completion of radiation (clinically relevant *). Functioning scores recovered 4 weeks after receiving chemotherapy **Symptoms:** Fatigue, nausea, and vomiting were significantly worse after completing radiotherapy (clinically relevant *). Four weeks after chemotherapy, complaints of fatigue remained.	No functioning or symptom scores differed statistically from baseline at one year.
		(QLQ-C30)		
**Kinami 2020**	Post-gastrectomy syndrome/ HRQOL > 1 year after partial gastrectomy (early vs. advanced stages)	Post-surgery (>1 year)		**Symptoms:** Dumping subscale was scored worse in patients with less stomach remnant. **Living status:** No differences in living status were observed between early and advanced gastric cancer patients. **HRQOL:** No difference in QOL between early and advanced gastric cancer patients.
		(PGSAS-45)		
**Munene 2012**	HRQOL after gastrectomy	Prior to surgery Post-surgery at: +3, +6, +9, +12, +15, +18, +21, +24, +30, +36, +42 months	**HRQOL:** HRQOL is deteriorated 3 months after gastrectomy for both total and partial gastrectomy.	**HRQOL:** HRQOL recovered after 6 months, for both total gastrectomy and partial gastrectomy.
		(FACT-G, FACT-Ga)		
**Park 2020**	HRQOL before and after total gastrectomy (TG) and distal gastrectomy (DG)	Prior to surgery Post-surgery at: +1, +2, +3 years		**Functioning:** Physical and role functioning were significantly worse in the TG group compared to the DG group after 2 and 3 years. **Symptoms:** Complaints in pain, reflux, eating restrictions, and anxiety were reported significantly more often in the TG group at all timepoints post-surgery. Fatigue and body image were reported significantly worse in the TG group after 2 and 3 years. The TG group scored worse for taste at 1 and 3 years post-surgery.
		(QLQ-C30, QLQ- STO22)		
**Xia 2020**	HRQOL in patients with gastric cancer vs. healthy references	>1 year post diagnosis		**HRQOL:** The mean EQ-5D utility score is significantly lower than the healthy controls. As well as the mean EQ-VAS score. **Domains:** The proportion of patients with problems in all 5 dimensions (pain, anxiety, self-care, usual activities, and mobility) is higher in patients with locally advanced gastric cancer.
		(EQ-5D)		
**Zygogianni 2018**	HRQOL after chemoradiation, (anterior-posterior vs. multifield technique)	Post-chemoradiation	**Functioning:** Global QOL scored nearly twice as much in the three-dimensional multifield group. **Symptoms:** Appetite loss and diarrhea were significantly better in the three-dimensional multifield group.	
		(QLQ-C30)		

* Clinically relevant = at least a difference of 10 points was observed between two measured groups; ns = not significant.

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
