# Peer review of "Health-Related Quality of Life in Locally Advanced Gastric Cancer: A Systematic Review"

_cancers, 2021, doi:10.3390/cancers13235934_

Round 1

Reviewer 1 Report

In my opinion, the authors of the review article: “Health-related quality of life in locally advanced gastric cancer: a systematic review” undertook an interesting, not thoroughly explored subject to discuss. On the other hand, the article has many shortcomings which should be refined before potential acceptance of the manuscript for publication. The authors themselves write about some limitations of their work. One of them, in my opinion is that the study was based on small numbers of results (articles). However the authors write that their article is the first one in such subject. Some specific comments are presented below:

  1. In the text the authors write that their review is based on 10 articles, however in the Table 1 there are 11 articles – should be explained;
  2. Table 2 is presented in such a way that it is really very difficult to take proper data into consideration and take conclusion;
  3. Lines 111-112 :… a study population below 20 patients was considered high risk …” – what does it mean? – should be explained;
  4. Lines 237 – 240 – the authors write about not published data, however there is reference 35 as cited – should be explained;
  5. Due to limited studies in the subject undertaken by the authors it would be maybe good idea to discuss the results of the study with the results considering other cancers (e.g. colon cancer) if there are such studies. Generally, in my opinion, the discussion should be broaden.

Author Response

Reviewer 1:

“In my opinion, the authors of the review article: “Health-related quality of life in locally advanced gastric cancer: a systematic review” undertook an interesting, not thoroughly explored subject to discuss. On the other hand, the article has many shortcomings which should be refined before potential acceptance of the manuscript for publication. The authors themselves write about some limitations of their work. One of them, in my opinion is that the study was based on small numbers of results (articles). However the authors write that their article is the first one in such subject. Some specific comments are presented below:”

Response: We highly appreciate that the reviewer underlines the importance of this study. We could not agree more with the reviewer on the fact that more research has to be done regarding the quality of life of patients with locally advanced gastric cancer. With this review, we hope to point out the lack of information regarding this topic and inspire clinicians and researchers to pay more focus to this important subject. We fully agree upon his or her comment regarding the limitations of the study. Therefore, we have edited line 281-283 to: “First, there is a lack of high quality studies investigating the quality of life in patients with locally advanced gastric cancer. Therefore, we only included a very small number of articles, hindering the possibility to perform a pooled analysis.  

Concerning the additional comments, see below for the detailed answers regarding the comments.

1. “In the text the authors write that their review is based on 10 articles, however in the Table 1 there are 11 articles – should be explained;”

Response: We agree that this discrepancy might have caused some confusion. This discrepancy is caused by two articles (Goody et al. 2016 and Kassam et al. 2010) describing the same cohort. We tried to elucidate the difference in numbers in the text (lines 130-131) and in the footnote of Table 1, but this was obviously not clear enough. To prevent the contradiction in the text, we have now combined the two studies in Table 1.

2. “Table 2 is presented in such a way that it is really very difficult to take proper data into consideration and take conclusion;”

Response: We appreciate the reviewer signaling the incompleteness of Table 2. Apparently the heading, as well as half of the studies are missing in the reviewers manuscript. Therefore we have resolved this issue, which clarifies the table significantly. We have discussed about whether we should add an extra text field with conclusions in the table. However this would make the font smaller, without adding extra information, which in our opinion would not contribute to the table.      

3. “Lines 111-112 :… a study population below 20 patients was considered high risk …” – what does it mean? – should be explained;”

Response: We thank the reviewer for pointing out this unclarity. We considered studies with small sample sizes at high risk of bias, as their data may be due to chance. As a cut off value we have taken a sample size minimum of 20 patients. We have added our argumentation, by adding the following text:

Lines 120-121 “Additionally, a study population below 20 patients was considered to be at risk of bias based on sample size and therefore excluded from further analysis.”

4. “Lines 237 – 240 – the authors write about not published data, however there is reference 35 as cited – should be explained;”

Response: We appreciate the reviewer’s suggestion and adjusted the text to:

Lines 260-263: “The first results show that after postoperative chemo(radio)therapy, the chemotherapy group had significantly better physical functioning (p=0.020, ES=0.25) and less dysphagia (p=0.010, ES=0.35) compared to the chemoradiotherapy group[35].”

5. Due to limited studies in the subject undertaken by the authors it would be maybe good idea to discuss the results of the study with the results considering other cancers (e.g. colon cancer) if there are such studies. Generally, in my opinion, the discussion should be broaden.

Response: Following the reviewer’s suggestion, we have incorporated an extra paragraph in the discussion to show the overall similarities between oesophageal cancer and locally advanced gastric cancer.

Lines 264-280: “Despite the small number of articles describing HRQOL in locally advanced gastric cancer, this study shows a trend in HRQOL similar to other cancers, e.g. oesophageal cancer. Treatment of oesophageal cancer (including chemoradiotherapy, surgery and neoadjuvant therapy combined with surgery) also resulted in a short-term deterioration of HRQOL, specified as four months from start of therapy [15]. Similar to the studies included in this systematic review, a statistically and clinically significant increase in fatigue was found in patients receiving chemoradiotherapy as well as a significant decrease in physical, role and social functioning and increase in loss of appetite and diarrhoea in patients receiving surgery. Within the long-term, classified at 12 months, comparable results were found to the locally advanced gastric cancer patients. The functioning scores also recovered to baseline, whereas patients receiving chemoradiotherapy had persistent appetite loss and patients receiving surgery had significantly increased dyspnoea and diarrhoea. Patients receiving neoadjuvant therapy and surgery reported better short-term HRQOL, compared to other therapies, however no differences in HRQOL were found compared to surgery alone in the long-term. When long-term HRQOL in oesophageal cancer patients was compared to healthy controls, HRQOL remained impaired, which is similar to the results of Brenkman et al. and Xia et al [30, 36, 39]. The overall similarity between both cancers support the trend shown in the HRQOL of locally advanced gastric cancer as presented in this study.

Reviewer 2 Report

Lines 48-49 – please be more specific / briefly explain Intergroup 0116 trial, MAGIC trial, and FLOT4 trial (what exactly they consist of)

Lines 91-92 – please exlain “If the proportion of patients exceeded 20% stage Ia studies were excluded”

Lines 93-97: please explain how the 6 month limit was set

Lines 203-204: please explain how this statement is supported -  “We found that HRQOL was impaired until 6 months of follow-up but from 6 months onwards, HRQOL recovered to pretreatment levels in most patients”.

Lines 218-219: please explain if this assertion “So, HRQOL can be important not only after treatment but also during follow-up in monitoring late treatment effects” is supported by data from the literature or if it is a speculation / assumption.

Lines 222-224: please explain

In conclusion:

- clear, well-defined research questions

- I positively note the use of relevant, internationally validated research tools

- novelty: this paper is the first systematic review investigating HRQOL in locally advanced gastric cancer patients even if certain limitations of the study are recognized.

Author Response

Reviewer 2:

  1. Lines 48-49 – please be more specific / briefly explain Intergroup 0116 trial, MAGIC trial, and FLOT4 trial (what exactly they consist of):”

Response: In response to the reviewer’s comment, we added the regimens to this sentence:

Lines 47-52: “Current curative treatment options for locally advanced gastric cancer include surgery with postoperative chemoradiotherapy (adjuvant leucovorin + fluorouracil and 45 Gray ) based on the Intergroup 0116 trial and perioperative chemotherapy based on the MAGIC trial (epirubicin+cisplatin+fluorouracil) and FLOT4 (fluorouracil+leucovorin+oxaliplatin+docetaxel) trial in Western patients, while in Asia adjuvant chemotherapy is the preferred treatment [7-12]”

  1. “Lines 91-92 – please explain “If the proportion of patients exceeded 20% stage Ia studies were excluded

Response: We thank the reviewer for noticing the unclarity in the text. To clarify our incentives, we have modified the sentence to:

Lines 94-97: “Studies researching various stages of gastric cancer were excluded if the proportion of patients exceeded 20% stage Ia. Since this study investigated the impact of multimodal treatment in gastric cancer, whereas Ia is often treated less intensively.”

  1. Lines 93-97: please explain how the 6 month limit was set

Response: We are happy to explain how the limit was determined. The short-term effects needed to display the direct effect of treatment, whereas the long-term was supposed to reflect the late treatment effects. Additionally, due to the poor prognosis of locally advanced gastric cancer, the late treatment effects were at risk of being overshadowed by deterioration in HRQOL due to recurrence of disease. Therefore we based the limit of six months, similar to available literature in cancer types with poor prognosis (1). We have changed the text accordingly into:

Lines 103-105: “Based on available literature, short-term HRQOL was defined as HRQOL scores within 6 months after treatment[19].”

  • Toms C, Steffens D, Yeo D, Pulitano C, Sandroussi C. Quality of Life Instruments and Trajectories After Pancreatic Cancer Resection: A Systematic Review. Pancreas. 2021 Sep 1;50(8):1137-1153. doi: 10.1097/MPA.0000000000001896. PMID: 34714277.

  1. “Lines 203-204: please explain how this statement is supported  “We found that HRQOL was impaired until 6 months of follow-up but from 6 months onwards, HRQOL recovered to pretreatment levels in most patients”.

Response: We appreciate the suggestion. We have elaborated the statement, by adding the deteriorated subscales of HRQOL in the following paragraph:

Lines 221-225: “The impact of treatment is reflected as an impaired short-term HRQOL. Within studies researching the impact of surgery, a decline in physical, mental and functioning scores were seen from 6 weeks to 3 months after surgery. This pattern was also observed after chemoradiotherapy. However, the long-term impact (after 6 months) showed a recovery of HRQOL for both surgery and chemoradiotherapy [23-25, 34].”

  1. “Lines 218-219: please explain if this assertion “So, HRQOL can be important not only after treatment but also during follow-up in monitoring late treatment effects” is supported by data from the literature or if it is a speculation / assumption.”

Response: We understand the need for more explanation. This statement is based on the results presented by  Avery et al. and Munene et al. These studies found a deterioration in HRQOL at the time of recurrence. However, we fully agree that the choice of words is not fully in line with the proposed statement made. As a result of the reviewer’s question, we have adjusted the text towards:

Lines 237-239: “So, HRQOL can be important not only after treatment but also during follow-up in monitoring disease progression.”

  1. “Lines 222-224: please explain”

Response: We appreciate the reviewer’s suggestion to give more explanation regarding the statement made, hence we have changed the text to:

Lines 243-245: “Park et al. described worse long-term HRQOL in patients who underwent a total gastrectomy compared to distal gastrectomy, in both functioning (physical and role) as well as seven symptom scores (fatigue, pain, reflux, eating restrictions, anxiety, taste and body image)[35].”

  1. In conclusion:
  • clear, well-defined research questions
  • I positively note the use of relevant, internationally validated research tools
  • novelty: this paper is the first systematic review investigating HRQOL in locally advanced gastric cancer patients even if certain limitations of the study are recognized.

Response: We thank this reviewer for the complements and the comments that were made regarding our study. The input has improved the quality and clarity of the study.

Round 2

Reviewer 1 Report

I approve the authors' comments to my review. The manuscript was well corrected. However in my opinion Table 2 is still not very clear. Maybe it should be edited in more readible form?

I recomend the manuscript for publication.

Author Response

Thank you for reviewing our manuscript.

We added a new table 2 and removed the old one (track changes even made it more unclear). Sorry. We have changed the columns and the text. Hopefully you will find it more readable this way. I have also attached the new table 2 to review separately. 

Kind regards
